# OpenReview forum: "Reducing the Probability of Undesirable Outputs in Language Models Using Probabilistic Inference"
_NeurIPS.cc/2025/Conference — NeurIPS 2025 poster_

### Official Review · Reviewer_8tur · 2025-06-30

**Clarity:** 3
**Significance:** 1
**Originality:** 1
**Rating:** 2
**Confidence:** 3

**Summary:**

The paper introduces RePULSe (Reducing the Probability of Undesirable Low-reward Sequences), a novel method aimed at explicitly reducing the likelihood of undesirable outputs from LLMs. Unlike standard RL approaches such as REINFORCE that optimize for expected reward and indirectly reduce low-reward outputs, RePULSe uses different proposal and target distribution to explicitly penalize them. It achieves this by defining a target distribution $\sigma_\theta$ that amplifies the mass on undesirable outputs, and learning a proposal distribution $q_\xi$ to efficiently sample from it. Lastly, the unlikelihood-style loss term that directly reduces the probability of these sampled low‑reward outputs are added to standard RL loss term.
The authors evaluate RePULSe in experimental toxicity and red-teaming settings, demonstrating that it achieves a lower probability of undesirable outputs and improves adversarial robustness, without degrading expected reward performance compared to traditional RL baselines.

**Questions:**

- The use of SNIS with a learned proposal in high-dimensional language model output spaces raises concerns about estimator variance. Could the authors provide empirical analysis or theoretical discussion regarding the variance and reliability of SNIS in their setting? For a simpler approach, convergence plot for multiple experiments with different seeds would suffice
- Have the authors considered evaluating their approach on more recent open-source models (e.g., LLaMA3 or Gemma2) to better demonstrate generality?
- The adversarial robustness evaluation uses only 10 held-out prompts. Could the authors justify this choice and discuss whether a larger and more statistically robust evaluation was considered?
- To add the reliability of the findings, could author add some of the seminal methods which share concept of penalizing low-reward sequence such as RRHF, SLiC-HF or unlikelihood training?

**Ethical Concerns:**

["NO or VERY MINOR ethics concerns only"]

**Final Justification:**

The author didn't provide sufficient experimental/theoretical evidence for the rebuttal.
The author didn't generalize the finding with proposed comment, such as using newer or larger model, additional investigation, and larger prompt size, although some of the concerns might be able to easily addressed with the limited resource.
I recommend that the authors substantially revise and improve the manuscript and I will keep my score unchanged.

**Limitations:**

yes

**Paper Formatting Concerns:**

Minor issue in title. The paper's title ("Expected Reward Misses the Mark: Learning Guided
Proposals to Reduce Undesirable Language Model Outputs") is different from original title ("Reducing the Probability of Bad Outputs in Language Models Using Probabilistic Inference") and don't share core key word such as "Probabilistic Inference"

**Quality:**

2

**Strengths And Weaknesses:**

### Strengths

 - The paper addresses an important  issue in LM alignment, namely how to directly reduce the probability of undesirable outputs, rather than just optimizing for reward.
 - The paper is well-organized and conceptually clear. The separation between policy optimization ($p_\theta$) and proposal learning ( $q_\xi$) is logical and effectively executed.
 - The paper frames undesirable output suppression as probabilistic inference, with theoretically grounded approach
 - In both synthetic and real-data setups, RePULSe outperforms standard RL baselines in reducing bad output probability and improving adversarial robustness.

### Weaknesses

 - The method’s relevance is somewhat diminished by recent trends in alignment, such as Direct Preference Optimization (DPO), which do not involve online sampling from $p_\theta$. The initial motivation could be broadened to account for these developments. For example, there are papers that even in DPO, the probability of the bad sample can increase provided that the gap between preferred sample goes higher (e.g [1]). Modifying the motivation to includes such findings might make the contribution stronger and more general
 - The discussion related to the finding that the diversity of RLHF  outputs decreases significantly needs a reference (e.g. [2]), which is also related to  the construction of the proposal distribution  $q_\xi$ (mode-seeking property vs mass-covering property)
 - Key components of the method—separation between policy and proposal distribution, sampling from different target distribution to focus on bas samples, and applying unlikelihood style negative gradients —have been discussed in previous work (e.g., RRHF, unlikelihood training) and not new
 - The use of self-normalized importance sampling (SNIS) with a learned proposal distribution in high-dimensional LM output spaces raises concerns about estimator variance and reliability. This can amplify the notorious problem of online RL approach. Empirical analysis of variance or convergence behavior of SNIS would be needed
 - The models used (DistilGPT2, SmolLM-135M) are small and outdated, limiting the generality and timeliness of the results. Additional experiment with recent open-sourced model such as Llama3 or Gemma2 would be need to further validate the findings.
 - The evaluation of adversarial robustness rely on a very small number (i.e. 10) of held-out prompts, raising concerns about variance and statistical reliability.
 - The paper does not compare against unlikelihood training, which shares the conceptual goal of reducing undesirable outputs and could serve as a strong baseline. Other alignment method shares online sampling and penalizing low-reward sequence such as RRHF or SLiC-HF are not considered either.

[1] Rafailov, et al. "Scaling laws for reward model overoptimization in direct alignment algorithms."
[2] Go, et al. "Aligning language models with preferences through f-divergence minimization."

---

> ### Author Rebuttal · Authors · 2025-07-31
>
> Thank you for your review! We appreciate that you acknowledge our paper addresses an important issue in LM alignment, that you found our approach theoretically grounded, and our paper clear to read.
>
> We partially omit quotes below for length limits (but respond to the full quote).
>
> ---
> > The method’s relevance is somewhat diminished by recent trends in alignment, such as Direct Preference Optimization (DPO), [...] Modifying the motivation to includes such findings might make the contribution stronger and more general
>
> Thanks for the suggestion and the reference. We mentioned DPO in the related works section: “DPO-based methods train on a preference dataset and may not generalize to avoiding undesirable outputs that are not present or under-represented in the data” but agree that this could be discussed further in the motivation. We will modify the motivation to include more on this.
>
> Also, to our knowledge, while DPO has been gaining momentum, RLHF is still the dominant paradigm for language model alignment. In particular, there was a paper at ICML 2024 [1] that found PPO/RLHF generally superior to DPO. Thus, we argue improvements in the RLHF paradigm are still valuable.
>
> [1] Xu, Shusheng, et al. "Is DPO Superior to PPO for LLM Alignment? A Comprehensive Study." ICML 2024.
>
> ---
> > The discussion related to the finding that the diversity of RLHF outputs decreases significantly needs a reference (e.g. [2]), [...]
>
> Thanks, we will include a reference in the updated version.
>
> ---
> > Key components of the method—separation between policy and proposal distribution, sampling from different target distribution to focus on bas samples, and applying unlikelihood style negative gradients —have been discussed in previous work (e.g., RRHF, unlikelihood training) and not new
>
> We will make it more clear in the paper where the novelty in our work lies. The key novelty in our work is 1) a general framework for combining RL with unlearning-style losses for improving safety/alignment properties, and 2) in **automatically and dynamically** (adjusting as the policy learns) **identifying** undesirable outputs on which the negative gradients should be applied, and specifically, doing so with a learned proposal.
>
> Existing unlearning/unlikelihood works that apply unlikelihood-style negative gradients (discussed in Related Work section - Unlearning subsection) use a **fixed** dataset assumed to be **known in advance**, often curated by hand. In contrast, we dynamically find new undesirable outputs throughout the whole training process on which the negative gradients should be applied.
>
> Separation between policy and proposal and sampling from a distribution focused on bad samples are both very general concepts. We agree that they have been discussed in previous work, yet respectfully disagree about the novelty of our contributions. Our work applies these concepts in a novel manner; our specific idea of learning a proposal that samples from a target distribution biased towards bad samples for the purposes of RL-based alignment has not appeared in previous work (to the best of our knowledge).
>
> Regarding RRHF, we believe it is much closer to PPO than it is to our work. RRHF samples responses from any of: the initial model $p_0$, the learned model $p_\theta$, other LLMs, or human experts. None of these are explicitly focused on finding new harmful responses for subsequent training/unlearning in the way our learned proposal $q_{\xi}$ does.
>
> ---
> > The use of self-normalized importance sampling (SNIS) with a learned proposal distribution in high-dimensional LM output spaces raises concerns about estimator variance and reliability. [...] Empirical analysis of variance or convergence behavior of SNIS would be needed
>
> We agree that high estimator variance and unreliability are possible. However, we disagree that empirical analysis of convergence is needed. While we agree convergence would help from an understanding perspective, from a practical perspective, we care about final outcomes such as the probability of bad outputs, which we evaluate.
>
> Arguably, it is more problematic if we run an empirical convergence analysis and show that our estimator has converged, because then one could (rightly) argue that our experiments are too simple and not reflective of scaled-up settings where we likely wouldn’t get convergence! Therefore, we disagree with the premise that convergence is necessary. The $q_{\xi}$ is a useful tool to generate approximate samples; we care about whether $q_{\xi}$ has converged or how good this approximation is only to the extent that it affects the final outcome/things we care about, which we directly evaluate in the paper.
>
> ---
> > Could the authors provide empirical analysis or theoretical discussion regarding the variance and reliability of SNIS in their setting? For a simpler approach, convergence plot for multiple experiments with different seeds would suffice
>
> As abovementioned, we respectfully disagree with the premise that this is necessary. However, we agree it would be interesting for better understanding, so we would consider adding this in a subsequent revision. We are curious - how would you suggest evaluating convergence? We could use SMC/probabilistic bounds on the log partition function (such as in [2]), but we would only have access to lower bounds (upper bounds require an exact target sample, which we do not have access to in our setting).
>
> [2] Zhao, Stephen, et al. "Probabilistic Inference in Language Models via Twisted Sequential Monte Carlo." ICML 2024.
>
> ---
> > The models used (DistilGPT2, SmolLM-135M) are small and outdated [...]
>
> > Have the authors considered evaluating their approach on more recent open-source models (e.g., LLaMA3 or Gemma2) [...]?
>
> We agree that experiments with more recent and larger models would be a valuable addition to our study, and plan to prioritize this for future work. As you rightly point out, the models we used, DistilGPT2 and SmolLM-135M, are smaller. We chose them due to the constraints of our available academic resources. Training models is significantly more expensive than inference. We also carry out a hyperparameter search for each method we evaluate, and use 10 seeds for each method to get confidence intervals/estimates of statistical significance. Altogether, the compute requirements quickly become significant even for the smaller models that we used.
>
> We try to do good science, which is why we follow this rigorous process to reduce the chance our results are just luck, and to try to be as fair to baselines as possible. So while we acknowledge that the lack of scale is a downside, we also believe that this should not be a primary reason for rejecting; this further worsens inequality in favour of those who have more resources at their disposal. We hope you agree that our experiments on these smaller models still provide a solid and valuable contribution to the field.
>
> ---
> > The evaluation of adversarial robustness rely on a very small number (i.e. 10) of held-out prompts, raising concerns about variance and statistical reliability.
>
> > [...] Could the authors justify [...] and discuss whether a larger and more statistically robust evaluation was considered?
>
> We agree that ideally we would have much more prompts, but note that GCG is expensive to run, and we have limited compute.
>
> To demonstrate our good faith, we ran GCG evaluations on 10 additional prompts (20 total) for RePULSe and the best performing baseline ($p_\theta$ proposal) which had the lowest GCG attack success rate. We handpicked all prompts to try to cover distinct conceptual areas with minimal overlap. Below, we show attack success rates with 95% confidence intervals in brackets:
>
> **10 prompts (chart in paper):**
> - Green dot ($p_\theta$ proposal baseline): 0.100 [0.060, 0.140]
> - Blue cross (RePULSe): 0.050 [0.010, 0.090]
>
> **20 prompts:**
> - Green dot ($p_\theta$ proposal baseline): 0.110 [0.070, 0.150]
> - Blue cross (RePULSe): 0.065 [0.020, 0.120]
>
> Overall, the difference between 10 and 20 prompts is small, and the values under 20 prompts are well within the confidence intervals we provided under 10 prompts, which provides some evidence regarding the reliability of our estimates.
>
> ---
> > The paper does not compare against unlikelihood training, [...] RRHF or SLiC-HF are not considered either.
>
> > [...] could author add some of the seminal methods which share concept of penalizing low-reward sequence such as RRHF, SLiC-HF or unlikelihood training?
>
> As we state in our paper, $L_u$ is a choice, and unlikelihood could be used as the loss there if desired. Other unlearning methods could also be used for $L_u$; one should feel free to use whatever works best. Similarly, SLiC-HF (or any other RL method) can be used instead of REINFORCE for $L_r$. Our main contributions are in 1) our general framework combining $L_r$ and $L_u$ rather than any specific choice of $L_r$ or $L_u$, and 2) the use of $\sigma_\theta$, with approximate sampling aided by learned $q_\xi$, to automatically and dynamically generate samples on which unlikelihood/unlearning style losses can be applied.
>
> We picked REINFORCE as $L_r$ and compared against REINFORCE baselines, which we believe is fair. If we pick SLiC-HF as $L_r$, then we think comparing against SLiC-HF baselines is fair. Ideally, we would consider all sorts of choices for $L_r$, and do comparisons of all of them; we could do this in future work, but this would be very compute intensive and we are resource-constrained.
>
> Regarding the choice of $L_u$, we could compare our choice of gradient ascent versus something like unlikelihood or other unlearning methods, but these would all be variants of RePULSe. We don’t do this comparison because our baselines are the other RL-based methods. **Considering more choices of $L_u$ would only make RePULSe better relative to baselines (as we would just pick the best performing $L_u$).**
>
> ---
> > Minor issue in title. [...]
>
> Thanks, we will fix this.

---

> ### Comment · Reviewer_8tur · 2025-08-06
>
> Thank the authors for the rebuttal and clarification. At this point, I will keep my score unchanged.
>
> On the issue of unreliability
> The SNIS estimator is designed to approximate an unknown target parameter. The variance of the estimator can cause unexpected result in different context, limiting the generalizability and reliability of using the proposed method in other context.
> Since the authors emphasize repeated experiments with different random seeds, he convergence plot can be easily plotted and provided.
> A key benefit of small-scale experiments is its ability to perform detailed ablations and gain deeper insights into model behavior. It is unclear why the authors claim that including convergence analysis would make their method more problematic. This concern is neither theoretically nor empirically addressed.
>
> On model size limitations
> While resource constraints are understandable, DistilGPT is significantly outdated and smaller than models commonly used today. Findings derived from million-parameter models are difficult to generalize without supporting evidence. Even with extensive hyperparameter tuning and multiple seeds, conclusions drawn from such a small model are limited to million scale model and lack relevance in the current LLM trends
> At a minimum, the authors should have included experiments with a more recent model of similar size - such as Qwen-2.5 500M - and conducted basic scaling analyses to assess the consistency of their findings at larger scales.I  recommend that the authors substantially revise the manuscript.

---

> ### Author Response · Authors · 2025-08-06
>
> Thanks for your response!
>
> > It is unclear why the authors claim that including convergence analysis would make their method more problematic
>
> This is a misunderstanding of our response. Let us try to explain/rephrase our response better.
>
> We said:
> > Arguably, it is more problematic if we run an empirical convergence analysis and show that our estimator has converged, because then one could (rightly) argue that our experiments are too simple and not reflective of scaled-up settings where we likely wouldn’t get convergence!
>
> What we meant by this is that: suppose we tested convergence and found that our estimator has converged. What conclusions would one draw from this? One conclusion is that everything is working as expected - great! But another possible conclusion is that the environment we are testing in is too simple, as in many real world problems/scaled up settings we wouldn't expect to get convergence! This would be problematic, because it would demonstrate that our experiments are not reflective of realistic settings.
>
> We believe our estimators have not converged in our (non-toy) setting, that this is expected because the setting is not so easy, and furthermore, as demonstrated by our results, despite having no convergence guarantee, we can still achieve useful empirical results (average return vs. probability of bad outputs) compared to strong RL baselines!
>
> Furthermore, we asked:
> > [...] how would you suggest evaluating convergence? We could use SMC/probabilistic bounds on the log partition function (such as in [2]), but we would only have access to lower bounds (upper bounds require an exact target sample, which we do not have access to in our setting).
>
> because it is not at all clear how convergence should be evaluated. One might suggest ESS as a measure, but ESS is not a reliable estimator of convergence; for example, you could have a bunch of samples that each have extremely low probability under the target distribution. The ESS in this case would be high, even though the proposal is nowhere near converged towards the target distribution!
>
> We would have greatly appreciated a response to the above question, not only because it would help clarify what the reviewer's understanding of convergence is (and see whether it differs from our understanding, which is that convergence happens when the proposal $q_\xi$ = $\sigma_\theta$ (the target)), but also because it is not trivially easy to evaluate convergence as the reviewer suggests (unless the reviewer has a different definition of convergence, in which case we should discuss why that definition is the right one).
>
> Regarding model scale, we agree that using larger models, even if still relatively small scale, would be useful, and we are prioritizing this for future work. Still, we reiterate that:
> > [...] the compute requirements quickly become significant even for the smaller models that we used
>
> We would also have appreciated a response to the many other points we responded to, which we assumed were also important factors in the reviewer's original score. If these responses were not convincing, we would appreciate understanding why not, and would be happy to discuss further. If these responses were convincing, we believe they merit a re-evaluation of the original score.

---

> > ### Comment · Reviewer_8tur · 2025-08-08
> >
> > Thanks for your reply. I appreciate your work and your active comment, which I believe might strengthen your work. I believe my score is accurate, so I will keep as it is.

---

> ### Author Response · Authors · 2025-08-09
>
> Thanks for your response and appreciation. However, we are disappointed in the lack of response to our questions and lack of engagement and substantive responses to our points.

---

### Official Review · Reviewer_KgRL · 2025-07-02

**Clarity:** 2
**Significance:** 2
**Originality:** 2
**Rating:** 3
**Confidence:** 5

**Summary:**

This paper presents RePULSe, a new model training method designed to mitigate the generation of undesirable outputs in language models. RePULSe utilizes a separately learned proposal distribution $q_{\xi}$, trained with a mass-covering objective, to actively sample outputs from the policy's low-reward regions. Evaluated in an analytical toy experiment on DistilGPT2 and a more realistic environment using SmolLM-135M-Instruct, RePULSe is shown to convincingly reduce the probability of undesirable outputs below baseline levels while maintaining comparable average reward.

**Questions:**

1. The experimental setup allocates twice the training episodes to baselines to ensure equal computational cost. However, this experimental approach raises some underlying concerns. To provide a clearer view of the performance dynamics, could the authors plot the trajectories of each method in the space of Figure 3 and 4 as a function of training progress?

2. The paper mentions $q_{\xi}$ is a language model and Appendix B details its parameterization, but the specific architecture of $q_{\xi}$ is not explicitly stated in the experiments. Could the authors clarify if $q_{\xi}$ shares the same architecture and size as $p_{\theta}$ (SmolLM-135M-Instruct)?

3. The paper validates the method on a relatively small-scale setup. While the authors acknowledge in the limitations that scaling could be challenging, but: have the authors performed any preliminary experiments on, or could they discuss the foreseen serious problems  when applying RePULSe to: a) larger-scale models (e.g., 7B+), and b) different model architectures?

**Ethical Concerns:**

["NO or VERY MINOR ethics concerns only"]

**Final Justification:**

While the method is novel, its validation on a severely outdated architecture significantly weakens its real-world relevance and contribution. For these reasons, I will maintain my original score.

**Limitations:**

Yes

**Quality:**

3

**Strengths And Weaknesses:**

Strengths

- Significance & Originality: Addresses the critical problem of suppressing rare, undesirable outputs in LMs. The proposed method, RePULSe, which uses a learned proposal ($q_{\xi}$) to find failure modes, is highly novel.

- Methodological Quality: The core mechanism of using a separately learned proposal model trained with a mass-covering objective is conceptually sound and well-justified for finding diverse failure modes.

- Experimental Rigor: The evaluation is strong, featuring an analytical toy experiment for clear validation, a fair computational budget comparison, and crucial adversarial robustness tests (GCG) that demonstrate a fundamental improvement in safety.

Weaknesses

- Scalability: The method's effectiveness is only validated on a single, small-scale model (135M parameters). Its scalability to various types of large models (e.g., 7B+), and the exploration efficiency of $q_{\xi}$ in a vastly larger output space, remain to be confirmed.

- Dependence on Reward Model Quality: The framework's performance appears to heavily depend on the quality of the reward model ($r$). The paper does not analyze the method's robustness to a lower-quality or biased reward model, nor does it discuss whether the interaction with such a model could introduce instability into the training process.

---

> ### Author Rebuttal · Authors · 2025-07-31
>
> Thank you for your review! We appreciate that you find our method highly novel, addressing a critical problem, conceptually sound, well-justified, and with strong evaluations.
>
> ---
> > Scalability: The method's effectiveness is only validated on a single, small-scale model (135M parameters). Its scalability to various types of large models (e.g., 7B+), and the exploration efficiency of $q_{\xi}$ in a vastly larger output space, remain to be confirmed.
>
> We agree that incorporating experiments with larger models would be a valuable addition to our study, and plan to prioritize this for future work. We actually tested two models, DistilGPT2 and SmolLM-135M, but indeed they are small-scale. We chose them due to the constraints of our available academic resources. Training models is significantly more expensive than inference. We also carry out a hyperparameter search for each method we evaluate, and use 10 seeds for each method to get confidence intervals/estimates of statistical significance. Altogether, the compute requirements quickly become significant even for the smaller models that we used.
>
> We try to do good science, which is why we follow this rigorous process to reduce the chance our results are just luck, and to try to be as fair to baselines as possible. So while we acknowledge that the lack of scale is a downside and that larger-scale results remain to be confirmed, we also believe that this should not be a primary reason for rejecting; this further worsens inequality in favour of those who have more resources at their disposal. We hope you agree that our experiments on these smaller models still provide a solid and valuable contribution to the field.
>
> ---
> > Dependence on Reward Model Quality: The framework's performance appears to heavily depend on the quality of the reward model ($r$). The paper does not analyze the method's robustness to a lower-quality or biased reward model, nor does it discuss whether the interaction with such a model could introduce instability into the training process.
>
> Firstly, note that RePULSe is a modified RL alignment method. For policy $p_\theta$, all RL methods (including well-established methods such as PPO in RLHF) are highly dependent on the reward model quality, and subject to reward hacking. Part of the reason we have a KL divergence penalty to the initial $p_0$ on all methods is to mitigate this. Furthermore, we evaluate using the same reward model for all methods, allowing for a fair comparison.
>
> You might also be referring to the fact that the learned proposal $q_{\xi}$ may reward hack, by finding low-reward sequences that are not actually bad/undesirable outputs, which might uniquely affect RePULSe and not other RL baselines. A few points here:
> 1. In our definition of the target distribution, we include (the current) $p_{\theta}$ as the prior, and we have a temperature parameter $\beta$ as well. This serves as a form of regularization (we are happy to discuss the technical details of this if desired) and helps mitigate reward hacking with some very specific sequences that the policy is very unlikely to produce but achieve low reward.
> 2. Even if such reward hacking occurs, note that the proposal $q_{\xi}$ samples are only used for $L_u$, on which we reduce their probability. Thus, while $q_{\xi}$ reward hacking might reduce the effectiveness of RePULSe in reducing the probability of the most bad outputs, it generally has no impact on the average case performance and does not introduce instability for the policy $p_{\theta}$.
>
> ---
> > The experimental setup allocates twice the training episodes to baselines to ensure equal computational cost. However, this experimental approach raises some underlying concerns. To provide a clearer view of the performance dynamics, could the authors plot the trajectories of each method in the space of Figure 3 and 4 as a function of training progress?
>
>
> We specifically allocated additional training to baselines to provide a fair comparison at roughly equal total computation; the additional training time given to baselines should only make the baselines stronger. To validate this and help address your concerns, we re-ran experiments with equal training episodes (all methods use 2 epochs) on the baselines compared to RePULSe. Results are below, with 95% confidence intervals in brackets (we apologize that the new rebuttal format prevents images):
>
> **Probability of bad outputs (lower is better):**
>
> - PPO 4 epochs (orange diamond in paper): 0.00017 [0.00005, 0.00034]
> - PPO 2 epochs: 0.00011 [0.00005, 0.00019]
>
> - REINFORCE 4 epochs (red x in paper): 0.00011 [0.00007, 0.00016]
> - REINFORCE 2 epochs: X = 0.00009 [0.00003, 0.00018]
>
> - REINFORCE with reward transform 4 epochs (purple triangle in paper): 0.00002 [0.00000, 0.00005]
> - REINFORCE with reward transform 2 epochs: 0.00006 [0.00002, 0.00011]
>
> - $p_\theta$ Proposal 4 epochs (green dot in paper): 0.00008 [0.00004, 0.00013]
> - $p_\theta$ Proposal 2 epochs: 0.00016 [0.00005, 0.00032]
>
> - RePULSe 2 epochs (blue plus in paper): 0.00004 [0.00002, 0.00006]
>
> **GCG attack success rate (lower is better):**
>
> - PPO 4 epochs (orange diamond in paper): 0.17 [0.09, 0.27]
> - PPO 2 epochs: 0.41 [0.33, 0.49]
>
> - REINFORCE 4 epochs (red x in paper): 0.17 [0.11, 0.24]
> - REINFORCE 2 epochs: X = 0.28 [0.20, 0.35]
>
> - REINFORCE with reward transform 4 epochs (purple triangle in paper): 0.22 [0.10, 0.35]
> - REINFORCE with reward transform 2 epochs:  0.27 [0.19, 0.34]
>
> - $p_\theta$ Proposal 4 epochs (green dot in paper): 0.10 [0.06, 0.14]
> - $p_\theta$ Proposal 2 epochs:  0.24 [0.11, 0.40]
>
> - RePULSe 2 epochs (blue plus in paper):  0.05 [0.01, 0.09]
>
> **Average return (higher is better):**
>
> - PPO 4 epochs (orange diamond in paper): -0.178 [-0.532, 0.165]
> - PPO 2 epochs: -0.445 [-0.753, -0.142]
>
> - REINFORCE 4 epochs (red x in paper): 0.010 [-0.371, 0.342]
> - REINFORCE 2 epochs: X = -0.207 [-0.544, 0.098]
>
> - REINFORCE with reward transform 4 epochs (purple triangle in paper): -0.149 [-0.531, 0.183]
> - REINFORCE with reward transform 2 epochs: -0.468 [-0.760, -0.208]
>
> - $p_\theta$ Proposal 4 epochs (green dot in paper): 0.127 [-0.283, 0.448]
> - $p_\theta$ Proposal 2 epochs: -0.122 [-0.471, 0.154]
>
> - RePULSe 2 epochs (blue plus in paper): -0.166 [-0.490, 0.130]
>
> In general, the 2 epochs comparisons make RePULSe look better than in the 4 epoch comparisons. On the return vs. probability of bad outputs plot, looking at only 2 epochs makes RePULSe clearly stand out on the bottom right, which is an improvement on the baselines’ Pareto frontier. The GCG evaluation is even more clearly in favor of RePULSe, with all 2-epoch baselines significantly worse than 4-epoch baselines. Overall, this highlights how impressive it is that we can still improve over baselines even when the baselines have **double** the policy updates!
>
> Unfortunately, plotting the full trajectories is expensive because of the sample-based probability of bad outputs evaluation (and GCG evaluation is also expensive), but we hope you find this additional evidence convincing.
>
> ---
> > The paper mentions is a language model and Appendix B details its parameterization, but the specific architecture of is not explicitly stated in the experiments. Could the authors clarify if shares the same architecture and size as (SmolLM-135M-Instruct)?
>
> Yes, it does. We will clarify that in a revision.
>
> ---
> > The paper validates the method on a relatively small-scale setup. While the authors acknowledge in the limitations that scaling could be challenging, but: have the authors performed any preliminary experiments on, or could they discuss the foreseen serious problems when applying RePULSe to: a) larger-scale models (e.g., 7B+), and b) different model architectures?
>
>
> For a), exploration might be a challenge, as we mentioned in our limitations section. But we also think there are improvements to be made that can mitigate this (this will be future work). Regarding b), we don’t expect any problems as our methods are general and can be applied to any model parameterization.

---

> > ### Comment · Reviewer_KgRL · 2025-08-06
> >
> > Thank you for the rebuttal and the additional experiments. The new 2-epoch training comparison is helpful in clarifying the method's performance against baselines given an equal number of updates.
> > However, my most critical concern remains unaddressed: the severe insufficiency of the model scale. This is not a technical limitation of computational resources, but a fundamental issue of the external validity of the paper's conclusions. The 135M-parameter model used is qualitatively different in its architecture and behavior from the modern large language models (e.g., 7B+) that are the focus of current safety research. Therefore, it is highly uncertain whether the safety conclusions drawn from this outdated model can be generalized.
> > In conclusion, while the method is novel, its validation on a severely outdated architecture significantly weakens its real-world relevance and contribution. For these reasons, I will maintain my original score.

---

### Official Review · Reviewer_ySPC · 2025-07-05

**Clarity:** 4
**Significance:** 2
**Originality:** 3
**Rating:** 4
**Confidence:** 4

**Summary:**

This paper introduces RePULSe, a new training method designed to more effectively reduce undesirable outputs from language models. The authors argue that standard reinforcement learning alignment methods fail to eliminate rare but harmful outputs because, as the model improves, it samples these "bad" sequences too infrequently to "unlearn" from them. RePULSe addresses this by augmenting the standard RL loss with an additional objective: it uses a learned "proposal" model to actively find and sample low-reward outputs, and then directly reduces their probability in the main LM. Experimental results show that, compared to standard RL baselines like PPO and REINFORCE, RePULSe significantly lowers the probability of bad outputs and enhances adversarial robustness, all while maintaining a similar level of overall expected reward.

**Questions:**

1. The hyperparameter $\alpha$ in Equation 2 balances the standard RL objective with the "undesirable output reduction" objective. How did you tune this, and how sensitive is the model's final performance (both reward and safety) to this value?
2. Does RePULSe require double the sampling computation per step compared to baselines? How would this affect training time?

**Ethical Concerns:**

["NO or VERY MINOR ethics concerns only"]

**Final Justification:**

I'll be keeping my positive score. I still think the core challenge in much of RL is the exploration problem, but I see how the author's work effectively tackles the goal of safety alignment by cleaning up failure modes.

**Limitations:**

See above.

**Quality:**

3

**Strengths And Weaknesses:**

The paper is clearly written, with a logical flow from motivation to method to results. The connection to related fields like probabilistic inference, unlearning, and adversarial training is well-articulated.

However, the paper's focus on eliminating low-reward outputs may not address the primary challenge in many current RL tasks, which is the exploration problem of finding rare, high-reward trajectories. For complex capability-driven tasks (e.g., reasoning), the main difficulty is not avoiding bad outputs (which is a vast space) but discovering the extremely sparse "good" outputs. RePULSe is framed for safety alignment, assuming that standard RL is sufficient for finding good solutions, and its primary contribution is cleaning up the failure modes. This makes it highly relevant for safety, but potentially less so for tasks where peak performance is the bottleneck.

The experiments, while well-designed, are conducted on a relatively limited scale, which may affect the generalizability of the findings. The primary experiments use a small model (SmolLM-135M), a short generation length (20 tokens), and a modest-sized dataset (20,000 prompts). While this serves as an effective proof-of-concept, it is an open question how the dynamics of RePULSe would play out with state-of-the-art, multi-billion parameter models and more complex, long-form generation tasks.

---

> ### Author Rebuttal · Authors · 2025-07-31
>
> Thank you for your review! We appreciate that you find our method highly relevant for safety (which is our main focus), our experiments well-designed, and our paper well-written and articulating well the connection to related fields.
>
> ---
> > However, the paper's focus on eliminating low-reward outputs may not address the primary challenge in many current RL tasks, which is the exploration problem of finding rare, high-reward trajectories. For complex capability-driven tasks (e.g., reasoning), the main difficulty is not avoiding bad outputs (which is a vast space) but discovering the extremely sparse "good" outputs. RePULSe is framed for safety alignment, assuming that standard RL is sufficient for finding good solutions, and its primary contribution is cleaning up the failure modes. This makes it highly relevant for safety, but potentially less so for tasks where peak performance is the bottleneck.
>
> We generally agree with this assessment, but we don’t think this should count as a weakness. We explicitly show that we can roughly maintain RL (average case/standard sampling) performance while improving safety properties. A method that improves one (important) property while keeping other properties constant is definitely valuable.
>
> ---
> > The experiments, while well-designed, are conducted on a relatively limited scale, which may affect the generalizability of the findings. The primary experiments use a small model (SmolLM-135M), a short generation length (20 tokens), and a modest-sized dataset (20,000 prompts). While this serves as an effective proof-of-concept, it is an open question how the dynamics of RePULSe would play out with state-of-the-art, multi-billion parameter models and more complex, long-form generation tasks.
>
> We agree that incorporating experiments with larger models would be a valuable addition to our study, and plan to prioritize this for future work. Indeed, the models we tested (DistilGPT2 and SmolLM-135M) are small-scale. We chose them due to the constraints of our available academic resources. Training models is significantly more expensive than inference. We also carry out a hyperparameter search for each method we evaluate, and use 10 seeds for each method to get confidence intervals/estimates of statistical significance. Altogether, the compute requirements quickly become significant even for the smaller models that we used.
>
> We try to do good science, which is why we follow this rigorous process to reduce the chance our results are just luck, and to try to be as fair to baselines as possible. So while we acknowledge that the lack of scale is a downside and that larger-scale results remain to be confirmed, we hope you agree that our experiments on these smaller models still provide a solid and valuable contribution to the field.
>
> ---
> > The hyperparameter in Equation 2 balances the standard RL objective with the "undesirable output reduction" objective. How did you tune this, and how sensitive is the model's final performance (both reward and safety) to this value?
>
> We used a simple coarse binary-search-like approach (e.g., try 0.001, 0.01, 0.1, 1, then say 0.01 and 0.1 work best, then try 0.03 as well). We didn’t do an exhaustive search, limiting ourselves to approximately logarithmic/exponential intervals of roughly 3x increments. The model’s performance does depend on this value, but given that we didn’t do a very exhaustive search (and our intervals are fairly coarse), we think the sensitivity isn’t too much of a problem.
>
> ---
> > Does RePULSe require double the sampling computation per step compared to baselines? How would this affect training time?
>
> Yes, this roughly doubles the training time for each policy ($p_{\theta}$) update. We explicitly account for this in our experiments by allocating double the number of update steps for $p_{\theta}$ to baselines (which gives roughly equal total computation and wall-clock time) to keep comparisons fair. Thus, the baselines in our paper are significantly stronger than they would be if we gave the same number of $p_{\theta}$ updates (see our response to Reviewer KgRL if interested in evidence for this), and highlights how impressive it is that we can still improve over baselines even when the baselines have double the policy updates!

---

> ### Comment · Reviewer_ySPC · 2025-08-05
>
> Thank the authors for the rebuttal and clarification. I believe my current score is an accurate reflection of the contributions of this work.

---

### Official Review · Reviewer_ujiK · 2025-07-23

**Clarity:** 4
**Significance:** 3
**Originality:** 2
**Rating:** 4
**Confidence:** 3

**Summary:**

Authors propose RePULSE, a training method that aims at reducing the probability of bad sequences.

This is being done in the two steps:
1) Find undesirable outputs: It uses probabilistic inference methods to draw low-reward samples more consistently as one of the limitation with standard RL approaches policies get better at sampling high reward outputs and naturally does not sample low-reward outputs.
   * A target distribution \sigma_{\theta} amplifies the low reward regions of the current LM policy \p_{\theta}, and a learned proposal q_{\xi} that aids in sampling from \sigma_{\theta}.
   * it is learned simultaneously while optimizing \p_{\theta}.
   * \sigma_{\theta} uses SNIS based on learned proposal initialized from \p_{\theta}
   * \p_{\theta} != q_{\xi}

2) Reducing the probability of the bad outputs from 1: This is achieved by using gradient ascent to both maximize the expected reward and directly suppress the probability of undesirable sequences by adding a second loss term that penalizes the low-reward samples.

Baselines: PPO, REINFORCE, Transformed REINFORCE,  \p_{\theta} as learned proposal q_{\xi}
Experiments:
   * Toy experiment: LM: DistilGPT2 , RM: Correa with T=2.
   * Realistic experiment: LM: SmolLL-135M-Instruct, RM: Deberta-v3-large-v2

**Questions:**

* Paper emphasizes on p\_{\theta} != q\_{\xi}, how would using an older version of p\_{\theta} as q\_{\xi} impact. There could be benefits of this approach computationally.

**Ethical Concerns:**

["NO or VERY MINOR ethics concerns only"]

**Final Justification:**

I thank the authors for the clarifications and agree with the novelty in the proposed approach. As also mentioned by other reviewers and the discussions with authors, further analysis into the approach's scalability as model size varies is essential to further strengthen the contributions as the analysis performed by authors include DistilGPT2 and SmolLM-135M which are smaller scale model and this poses a question mark whether this approach will scale well. But the novel approach proposed by the authors demonstrating positive results should also be considered and this has majorly influenced my rating.

**Limitations:**

yes the paper addresses major limitations but some additional limitations can be addressed based on the points mentioned under weaknesses and questions section.

**Paper Formatting Concerns:**

No concerns with formatting.

**Quality:**

3

**Strengths And Weaknesses:**

Strengths:
* Overall: Paper is very well written, clearly indicates the problem and motivation for the same.
* RePULSE correctly identifies and tries to tackle one of the limitations of the RL alignment based approaches by proposing to learn a proposal model q\_{\xi} is incentivized to explore low reward regions of the distribution in order to reduce the probability of bad sequences.
* Choice to train q\_{\xi} by minimizing Mass covering objective leads to more robustness, as that encourages q\_{\xi} to find bad sample sequences leading to more robust p\_{\theta}
* Strong baselines: The baselines being used as part of the experiments are fair and clearly justified in the paper. Use of Transformed REINFORCE as baseline challenges the suggested approach and similarly p\_{\theta} as one of the baselines helps in proving the significance of learned proposal q\_{\xi} as part of REPULSE.

Weaknesses:
* Authors mention that any q\_{\xi} can be used, but more ablations should be included to support this claim. Also the relationship between the learned proposal model and policy could have been studied better in terms of size/ params affecting RePULSE effectiveness.
* Paper claims REINFORCE has high variance but doesn't analyze how the RePULSe gradient affects the variance.
* Proposal model q\_{\xi} is explicitly optimized over samples that receive low score from the reward model, r. q\_{\xi} may end up learning/ exploiting hacks in the reward model. this may defeat the purpose of updating p\_{\theta} based on q\_{\xi}.
* RePULSE proposes co-training p\_{\theta} and q\_{\xi} that doubles model complexity/ training time, introduces new hyperparameters that brings in additional overhead of tuning these.

---

> ### Author Rebuttal · Authors · 2025-07-31
>
> Thank you for your review! We appreciate that you find our method correctly identifies and tries to tackle a limitation of RL alignment based approaches, our baselines fair and justified, and our paper well-written.
>
> ---
> > Authors mention that any q_{\xi} can be used, but more ablations should be included to support this claim.
>
> We mean to point out that one can use any $q_{\xi}$ in the general framework; this is a direct property of importance sampling/SMC that any proposal can be used. We are not making claims about which $q_{\xi}$ works best; some will likely work better than others, and one should choose whichever works best. We consider all choices of learned $q_{\xi}$ as variations of RePULSe, so exploring the choice of $q_{\xi}$ could only make our results stronger (as we would pick the best performing $q_{\xi}$).
>
> ---
> > Also the relationship between the learned proposal model and policy could have been studied better in terms of size/ params affecting RePULSE effectiveness.
>
> We agree that this would be interesting to see, and plan to investigate in future work. Unfortunately, increasing size/params is difficult for us due to the constraints of our available academic resources. Training models is significantly more expensive than inference. We also carry out a hyperparameter search for each method we evaluate, and use 10 seeds for each method to get confidence intervals/estimates of statistical significance. Altogether, the compute requirements quickly become significant even for the smaller models that we used. We try to do good science, which is why we follow this rigorous process to reduce the chance our results are just luck, and to try to be as fair to baselines as possible. So while we acknowledge that we could have further studied different sizes, we hope you agree that our experiments still provide a solid and valuable contribution to the field.
>
> ---
> > Paper claims REINFORCE has high variance but doesn't analyze how the RePULSe gradient affects the variance.
>
> Are you referring to the discussion in the background section, e.g., line 81? Here, we are just paraphrasing prior work, and these comments about variance reduction through baselines apply to all policy gradient RL methods, not just RePULSe specifically.
>
> ---
> > Proposal model q_{\xi} is explicitly optimized over samples that receive low score from the reward model, r. q_{\xi} may end up learning/ exploiting hacks in the reward model. this may defeat the purpose of updating p_{\theta} based on q_{\xi}.
>
> Indeed, it is possible $q_{\xi}$ could reward hack, finding low-reward sequences that are not actually bad/undesirable outputs. Note however that in our definition of the target distribution, we include (the current) $p_{\theta}$ as the prior, and we have a temperature parameter $\beta$ as well. This serves as a form of regularization (we are happy to discuss the technical details of this if desired) and helps mitigate reward hacking with some very specific sequences that the policy is very unlikely to produce but achieve low reward.
>
> ---
> > RePULSE proposes co-training p_{\theta} and q_{\xi} that doubles model complexity/ training time, introduces new hyperparameters that brings in additional overhead of tuning these.
>
> Regarding doubling, we keep the comparison vs. baselines fair by giving baselines double the updates on the policy $p_{\theta}$, which leads to roughly similar total computation/wall-clock time. This makes the baselines in our paper very strong (see our response to Reviewer KgRL for what happens if we match policy updates), and highlights how impressive it is that we can still improve over baselines even when the baselines have double the policy updates!
>
> ---
> > Paper emphasizes on p_{\theta} != q_{\xi}, how would using an older version of p_{\theta} as q_{\xi} impact. There could be benefits of this approach computationally.
>
> Using an older version of $p_{\theta}$ (instead of the current $p_{\theta}$ we use in our baseline) is an interesting idea - thanks for the suggestion! We suspect that it wouldn’t help because our target distribution is based on the current $p_{\theta}$ (and some experiments we did using the original $p_{\theta}$ in the target didn’t perform well), so the current $p_{\theta}$ should generally be closer to the target than the older $p_{\theta}$. But it could be possible that the older $p_{\theta}$ has more low-reward sequences, which might have higher probability mass under the new target distribution. We could try this in future revisions.

---

> > ### Comment · Reviewer_ujiK · 2025-08-07
> >
> > I thank the authors for the clarifications and agree with the novelty in the proposed approach. Nevertheless, further analysis into the approach's scalability as model size varies is essential to further strengthen the contributions and thus I'd like to maintain the current ratings.

---

### Decision · Program_Chairs · 2025-09-17

**Decision:**

Accept (poster)

**Comment:**

My recommendation is to accept the paper.

The paper proposes a method for reducing the probability of undesirable outputs from language models further than conventional RL approaches. The method samples undesirable outputs from the support of the policy model (e.g., by targeting samples that fall in the lower tail of the reward distribution, or samples that result from tilting the policy distribution to favor low-reward samples) and adds an "unlikelihood" term to the policy update gradient to penalize such outputs. The authors show that the method behaves as expected on a mixture of non-adversarial and adversarial prompts, as well as a toy example and a small adversarial robustness setting.

The reviewers agreed that the method used a novel mixture of ideas in an RL setting to deal with safety issues where the reduction in probability of bad outputs may not be sufficient. They agreed that in the evaluation setting, it appeared that the method was successful, including against a strong baseline that was motivated by the approach itself.

The primary criticism was that the evaluation was too limited, especially with respect to policy model architecture. This was a unanimous concern, but because the idea in the paper is novel and the experiments are well-designed to demonstrate the conceptual underpinnings and mechanisms of the method, I am exercising my own judgment to overrule that concern. It is true that actually applying this method to larger scale models would be useful, but the point of this paper is not to propose a SOTA method, but to introduce a new approach to a specific problem, which it does well.

Some other concerns were about the convergence properties of the method. I found this concern less compelling; as the authors noted, the primary goal is not to estimate a particular quantity, but to achieve a targeted reduction in bad output probability without affecting the mean reward. However, some analysis of stability of the algorithm here would be useful for thinking about what the essential pieces of the approach are for applying to larger models.

I do think that the paper could potentially be strengthened by at least showing that the probability of bad outputs persists at a non-negligible level in larger scale models that have been safety-tuned, at least as an empirical example for motivation.